# MacDC: Masking-augmented Collaborative Domain Congregation for Multi-target Domain Adaptation in Semantic Segmentation

## Abstract

This paper addresses the challenges in multi-target domain adaptive segmentation which aims at learning a single model that adapts to multiple diverse target domains. Existing methods show limited performance as they only consider the difference in visual appearance (*style*) while ignoring the *contextual* variations among multiple target domains. In contrast, we propose a novel approach termed Masking-augmented Collaborative Domain Congregation (MacDC) to handle the *style gap* and *contextual gap* altogether. The proposed MacDC comprises two key parts: collaborative domain congregation (CDC) and multi-context masking consistency (MCMC). Our CDC handles the style and contextual gaps among target domains by data mixing, which generates image-level and region-level intermediate domains among target domains. To further strengthen contextual alignment, our MCMC applies a *masking*-based self-supervised augmentation consistency that enforces the model's understanding of diverse contexts together. MacDC directly learns a single model for multi-target domain adaptation without requiring multiple network training and subsequent distillation. Despite its simplicity, MacDC shows efficacy in mitigating the style and contextual gap among multiple target domains and demonstrates superior performance on multi-target domain adaptation for segmentation benchmarks compared to existing state-of-the-art approaches.

## 1 Introduction

Recent advancements in semantic segmentation owe much to large-scale datasets with manual annotations, which prove costly and time-intensive. One alternative is to utilize either existing labeled datasets (Cordts et al., 2016; Varma et al., 2019) or synthetic datasets (Richter et al., 2016; Ros et al., 2016) capable of automatic annotation generation. Nevertheless, the model trained on the source dataset often experiences a performance degradation when applied to the target dataset due to the *source-target gap*. Unsupervised domain adaptation (UDA) has been introduced to handle this issue in semantic segmentation, typically focusing on one target domain. The most recent UDA approaches employ self-supervised augmentation consistency, such as ClassMix (Tranheden et al., 2021), to align the source-target gap. Despite this, these UDA methods are tailored for scenarios with a single target domain, which can frequently be deviated from in practical applications. For instance, the model deployed in autonomous driving systems has to be generalized in diverse target domains. A significant gap arises within these domains, referred to as the *target-target gap*, due to different visual appearances and scene contexts caused by variations in weather, driving scenes, and lighting conditions.

Multi-target domain adaptation (MTDA) aims to adapt a single model to multiple target domains by mitigating the *source-target gap* and the *target-target gap* altogether. Compared to the extensive research in UDA, MTDA is less investigated with only a few studies on semantic segmentation (Saporta et al., 2021; Isobe et al., 2021; Lee et al., 2022; Zhang et al., 2023). Existing MTDA for semantic segmentation methods align the target-target gap by employing either explicit style transfer in image space (Isobe et al., 2021; Lee et al., 2022) or implicit style transfer in feature space (Zhang et al., 2023). However, the target-target gap does not solely arise from differences in visual appearance (*style*) but also from *contextual* variations that are currently ignored by existing MTDA

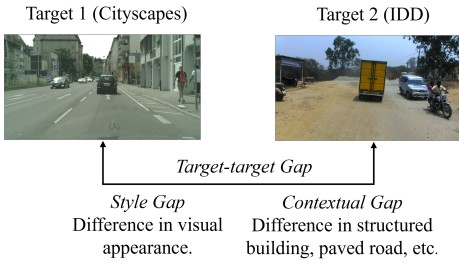

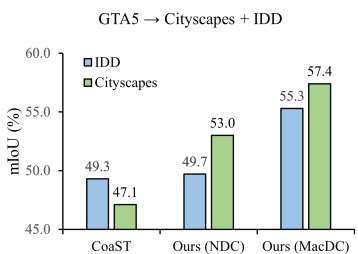

Figure 1: Illustration of the style gap and the contextual gap among the target domains.

Figure 2: The performance comparison of CoaST, NDC, and MacDC.

approaches. For instance, Fig. 1 illustrates two target images from the Cityscapes (Cordts et al., 2016) and the IDD dataset (Varma et al., 2019), respectively. The first image (from Cityscapes) is collected in an urban scene with organized structures and concrete roads, whereas the second image (from IDD) is collected in a rural scene with disorganized structures and unpaved roads. The difference in terms of the buildings and roads may result in a significant contextual gap among these images.

In this work, we present a novel MTDA framework named masking-augmented collaborative domain congregation (MacDC), which aims to mitigate the *style gap* and the *contextual gap* among the multiple target domains altogether. The proposed MacDC framework learns robust domain invariant features by utilizing *self-supervised augmentation consistency* on multiple target domains. It consists of two key components: collaborative domain congregation (CDC) and multi-context masking consistency (MCMC).

**CDC**. Collaborative domain Congregation (CDC) aims to align the *style gap* and the *contextual gap* by generating intermediate domains from the target domains. To bridge the style gap, CDC employs an image-level domain interpolation to enable the interchange of image styles for multiple target domains. Nevertheless, the image-level domain interpolation is unable to effectively handle the contextual gap among the target domains. To cope with it, CDC applies region-level domain interpolation to mingle diverse scene contexts from different target domains. The resulting images by the region-level domain interpolation facilitate the model in exploring a more diverse range of scene contexts.

**MCMC**. To effectively promote the alignment of the *contextual gap* among target domains, we present multi-context masking consistency (MCMC). Previously, Hoyer et al. (2023) utilizes *masking*-based self-supervised augmentation consistency to learn image contexts, by enforcing prediction consistency between a masked image and its corresponding complete image. The model is then prompted to make semantic predictions of the masked-out patches with reference to their surrounding context within the image. We extend it and propose MCMC for the interpolated images mingled with different scene contexts from the target domains. MCMC effectively reinforces the model's learning of the *diverse contexts* from different target domains *together*.

**Our Contribution**. (1) To the best of our knowledge, we are the first to handle the *contextual gap* among the multiple target domains. (2) We propose a novel MTDA framework called MacDC which contains two key parts namely CDC and MCMC, to effectively mitigate both the *style gap* and the *contextual gap* altogether for MTDA in semantic segmnetation. (3) Despite its simplicity, our MacDC outperforms existing state-of-the-art approaches on MTDA for semantic segmentation benchmarks.

## 2 RELATED WORK

**Unsupervised Domain Adaptation with Masking.** UDA has been extensively studied in various computer vision tasks such as image classification (Zhang et al., 2018; Prabhu et al., 2021; Pan et al., 2019), object detection (Wu et al., 2021; Li et al., 2022a;b), and semantic segmentation (Vu et al., 2019; Pan et al., 2020; Hoyer et al., 2022). Existing UDA approaches for semantic segmentation adopt adversarial training, self-training, and self-supervised augmentation consistency. In adver-

sarial training, the model is optimized with a discriminator to learn domain-invariant features by aligning the gap in the input space (Hoffman et al., 2018; Pan et al., 2022), feature space (Vu et al., 2019), and output space (Tsai et al., 2018; Pan et al., 2020). Self-training involves a training loop that relies on target pseudo-labels generated with confidence thresholds (Shin et al., 2020) or category prototypes (Zhang et al., 2021). To enhance the self-training stability, (Tranheden et al., 2021) present a *mixing*-based self-supervised augmentation consistency named ClassMix. (Hoyer et al., 2023) exploits a *masking*-based self-supervised augmentation consistency to improve the model's contextual understanding of the target domain. Nevertheless, these approaches are restricted to scenarios with a single target domain. In this paper, we consider the multi-target scenario, where the model is trained to adapt to multiple target domains.

**Multi-target Domain Adaptation (MTDA).** MTDA proposes to adapt a single model to multiple target domains altogether. In comparison to UDA, MTDA has received relatively little attention, with few studies on image classification (Yu et al., 2018; Nguyen-Meidine et al., 2021), object detection (Kiran et al., 2022), and semantic segmentation (Saporta et al., 2021; Isobe et al., 2021; Lee et al., 2022; Zhang et al., 2023). Currently, MTDA methods for semantic segmentation typically involve an initial stage of learning multiple teacher networks using adversarial training followed by distilling knowledge to a single student network. Saporta et al. (2021) introduce an MTDA framework that employs adversarial training on a generator and multiple discriminators to adapt to different target domains. In addition, Isobe et al. (2021) and Lee et al. (2022) combine adversarial training with explicit style transfer in the image space to align the style discrepancies among the target domains. Zhang et al. (2023) further presents implicit style transfer in feature space to improve training robustness. Nonetheless, existing MTDA approaches concentrate solely on the style gap while ignoring the contextual variations among multiple target domains. We present a new MTDA framework using self-supervised augmentation consistency to mitigate both the style gap and the contextual gap across target domains altogether.

**Domain Interpolation.** Compared to directly adapting to the target domain, existing UDA methods attempt to generate intermediate domains for a smoother adaptation process from the source domain to the target domain. Gong et al. (2019) proposes to generate a flow of numerous intermediate domains based on the image styles. Tranheden et al. (2021) employs a class-based data mixing technique to generate intermediate domains by mingling the data from the source and the target domain. Chen et al. (2022) adopts several data mixing techniques together to generate intermediate domains. However, these works are designed for domain interpolation between a source and a target domain. In this work, we apply domain interpolation to generate intermediate domains among multiple target domains, which is used to mitigate the style gap and the contextual gap in the MTDA setting.

## 3 METHODOLOGY

In this section, we present a new framework named masking-augmented collaborative domain congregation (MacDC) for MTDA in semantic segmentation. First, we introduce the MTDA setting (Sec. 3.1) and present naive domain congregation (Sec. 3.2). Then, we propose collaborative domain congregation to bridge the *style gap* and the *contextual gap* among target domains (Sec. 3.3). Furthermore, we present multi-contextual masking consistency to further mitigate the *contextual gap* in target domains (Sec. 3.4). Our loss function to optimize MacDC is shown in the last.

### 3.1 PROBLEM DEFINITION

The task of multi-target domain adaptation (MTDA) is learning a single model to adapt from a source domain to multiple target domains. Let the source domain be denoted as $\mathbb{S} = \{x_n^{\mathbb{S}}, y_n^{\mathbb{S}}\}_{n=1}^{N^{\mathbb{S}}}$ which contains $N^{\mathbb{S}}$ labeled instances. Each instance consists of an RGB image $x^{\mathbb{S}} \in \mathbb{R}^{H,W,3}$ and the corresponding one-hot ground truth label $y^{\mathbb{S}} \in \mathbb{B}^{H,W,C}$. In addition, there are $K$ unlabeled target domains $\{\mathbb{T}_1, \mathbb{T}_2, \ldots, \mathbb{T}_K\}$, where the target domain $\mathbb{T}_i = \{x_n^{\mathbb{T}_i}\}_{n=1}^{N^{\mathbb{T}_i}}$ contains $N^{\mathbb{T}_i}$ unlabeled images, and $x^{\mathbb{T}_i} \in \mathbb{R}^{H',W',3}$ denotes an RGB image from $\mathbb{T}_i$. Note that $C$ represents the total number of categories, $(H, W)$ and $(H', W')$ represent the height and the width of the source and the target images, respectively. The *target-target gap* exist among these target domains due to the style and contextual discrepancies.

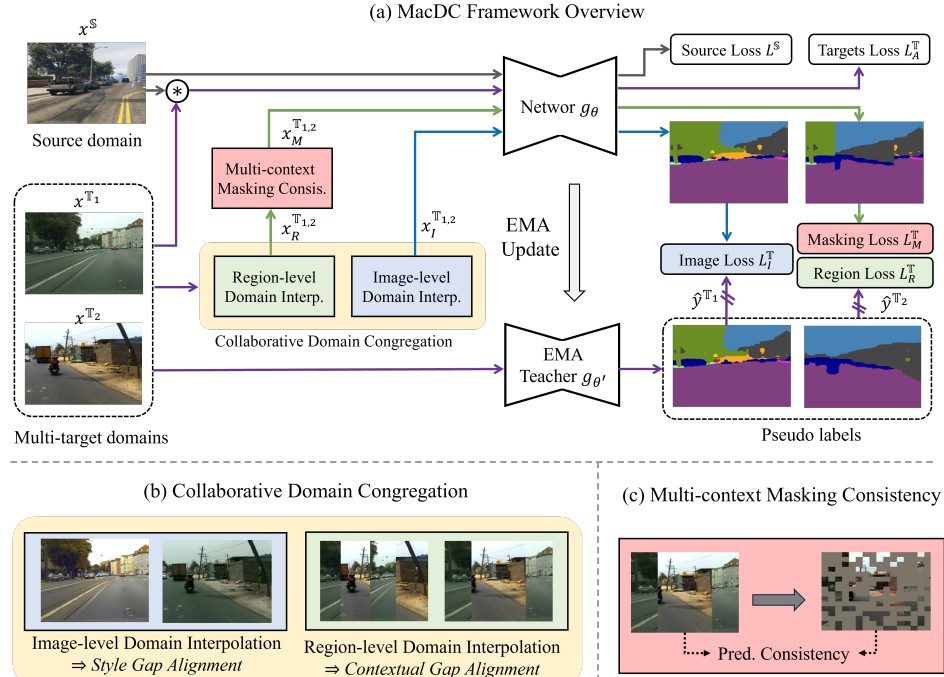

Figure 3: Overview of the proposed MacDC framework. Our MacDC framework consists of two novel parts: collaborative domain congregation and multi-context masking consistency. During training, the weights of the EMA teacher $g_{\theta'}$ get updated from the weights of the segmentation network $g_\theta$. After training, $g_\theta$ is used for evaluation on multiple target domains. Note that $\circledast$ is the operator of data augmentation using ClassMix.

## 3.2 NAIVE DOMAIN CONGREGATION

Given the unlabeled target domains $\{\mathbb{T}_1, \mathbb{T}_2, \ldots, \mathbb{T}_K\}$, naive domain congregation (NDC) is to simply merge the data of all the target domains together into a target *superdomain* $\mathbb{T}^\dagger$ which is formulated by

$$\mathbb{T}^\dagger = \bigcup_{i=1}^{K} \mathbb{T}_i. \tag{1}$$

By implementing NDC, the MTDA setting is transformed into a standard unsupervised domain adaptation (UDA) setting: one source and one target domain. Intuitively, it is possible to apply existing UDA approaches to learn a single model adapting from $\mathbb{S}$ to $\mathbb{T}^\dagger$. Tranheden et al. (2021) present a *mixing*-based self-supervised augmentation consistency, named ClassMix, as a simple and effective method for UDA in semantic segmentation. Inspired by this, we aim to directly learn a single MTDA model that can adapt from $\mathbb{S}$ to $\mathbb{T}^\dagger$ with ClassMix.

Given a source image and its corresponding label $\{x^\mathbb{S}, y^\mathbb{S}\}$, we train the segmentation network $g_\theta$ with a supervised loss $\mathcal{L}^\mathbb{S}$ defined by

$$\mathcal{L}^\mathbb{S} = \mathcal{C}(g_\theta(x^\mathbb{S}), y^\mathbb{S}),$$
$$\mathcal{C}(\hat{y}^\mathbb{S}, y^\mathbb{S}) = -\sum_{h,w}^{H,W} \sum_{c}^{C} y^\mathbb{S}_{(h,w,c)} \log(\hat{y}^\mathbb{S}_{(h,w,c)}), \tag{2}$$

where $\mathcal{C}$ represents the cross-entropy loss function. Given a randomly sampled image $x^{\mathbb{T}_i}$ from $\mathbb{T}_i$, we also generate it's semantic segmentation map $\hat{y}^{\mathbb{T}_i} = g_{\theta'}(x^{\mathbb{T}_i})$, where $g_{\theta'}$ is an EMA teacher of $g_\theta$ and its weights $\theta'$ were updated by $\theta$ with EMA (Tarvainen & Valpola, 2017). Then, we take $\{x^\mathbb{S}, y^\mathbb{S}, x^{\mathbb{T}_i}, \hat{y}^{\mathbb{T}_i}\}$ as the input and send them into *classmix* to generate the augmented image $x_A^{\mathbb{T}_i}$ and the corresponding pseudo-label $\hat{y}_A^{\mathbb{T}_i}$. These augmented images along with the corresponding

pseudo-labels are used to train $g_\theta$ with

$$\mathcal{L}_A^{\mathbb{T}_i} = \mathcal{F}\big(g_\theta(x_A^{\mathbb{T}_i})\big)\mathcal{C}\big(g_\theta(x_A^{\mathbb{T}_i}), \hat{y}_A^{\mathbb{T}_i}\big), \tag{3}$$

where $\mathcal{F}(\cdot)$ is to evaluate the quality of the segmentation predictions. Following Tranheden et al. (2021), $\mathcal{F}(\cdot)$ is defined by the ratio of pixel predictions surpassing a pre-defined threshold $\xi$

$$\mathcal{F}\big(y\big) = \frac{\sum_{h,w}^{H,W} \mathbb{1}\{\max_{c'} y_{(h,w,c')} > \xi\}}{H \cdot W}, \tag{4}$$

where $\mathbb{1}\{\cdot\}$ is an indicator function. The loss function $\mathcal{L}^\dagger$ for training NDC is denoted by

$$\mathcal{L}^\dagger = \mathcal{L}^{\mathbb{S}} + \lambda_A \sum_i^K \mathcal{L}_A^{\mathbb{T}_i}. \tag{5}$$

We compare the performance of NDC with the existing best MTDA for segmentation method CoaST (Zhang et al., 2023) under the MTDA benchmark: GTA5 → Cityscapes + IDD in Fig. 2. Despite its simpleness, NDC presents $49.7\%$ mIoU on IDD and $53.0\%$ mIoU on Cityscapes, outperforming CoaST on both two target domains shown in Fig. 2. However, there exists a performance gap between Cityscapes and IDD in NDC, as NDC does not consider the style and contextual discrepancies among the target domains (called the *style gap* and *contextual gap*). We postulate that mitigating the style and contextual gap would further contribute to the model's effective adaptation to multiple target domains.

### 3.3 COLLABORATIVE DOMAIN CONGREGATION

To bridge the *style gap* and the *contextual gap* among target domains, we present collaborative domain congregation (CDC) to connect target domains by generating intermediate domains among themselves. Specifically, our CDC incorporates image-level interpolation to bridge the *style gap* and region-level domain interpolation to alleviate the *contextual gap* simultaneously. On this basis, a target *superdomain* $\mathbb{T}^\ddagger$ is generated by assembling all the target domains and the generated intermediate domains.

**Image-level Domain Interpolation.** To bridge the *style gap*, our image-level domain interpolation adopts style transfer to create the intermediate domains by blending different target domain styles. Common style transfer methods for image-level domain interpolation are categorized into GAN-based methods (Lee et al., 2021; Zhu et al., 2017) and ColorTransfer (Reinhard et al., 2001). Regardless of the technical differences of these methods, we want to present the image-level domain interpolation in a uniform formulation. Suppose we have two samples $x^{\mathbb{T}_i}$ and $x^{\mathbb{T}_i}$ which are randomly selected from $\mathbb{T}_j$ and $\mathbb{T}_j$, our image-level domain interpolation is represented by

$$x_I^{\mathbb{T}_{i,j}} = G(x^{\mathbb{T}_i}, x^{\mathbb{T}_j}). \tag{6}$$

Here $G(x^{\mathbb{T}_i}, x^{\mathbb{T}_j})$ represents the translation function from $\mathbb{T}_i$ to $\mathbb{T}_j$, and $x_I^{\mathbb{T}_{i,j}}$ is the translated image that shares the content of $x^{\mathbb{T}_i}$ and the style of $x^{\mathbb{T}_j}$. Similarly, $x_I^{\mathbb{T}_{j,i}}$ is the translated image that shares the content of $x^{\mathbb{T}_j}$ and the style of $x^{\mathbb{T}_i}$. Based on the translated images, our image-level domain interpolation generates the image-level intermediate domains $\mathbb{T}_{i,j}^I = \{x_{I,n}^{\mathbb{T}_{i,j}}\}_{n=1}^{N^{\mathbb{T}_i}}$ and $\mathbb{T}_{j,i}^I = \{x_{I,n}^{\mathbb{T}_{j,i}}\}_{n=1}^{N^{\mathbb{T}_j}}$ between $\mathbb{T}_i$ and $\mathbb{T}_j$.

**Region-level Domain Interpolation.** In spite of the intermediate domains generated by image-level domain interpolation, a significant *target-target gap* still exists due to the contextual discrepancies from target domains. For example, one target domain contains urban street scenes with well-structured buildings and paved roads, while another target domain might include rural scenes with unstructured houses and dirt roads. To bridge the *contextual gap* among the target domains, our region-level domain interpolation generates the intermediate domains by utilizing data mixing on different target image regions. The common data mixing approaches for region-level domain interpolation include CowMix (French et al., 2020), FMix (Harris et al., 2020), and CutMix (Yun et al., 2019). Regardless of their technical differences, let $\mathbf{R}$ indicate a binary matrix designating which pixel to copy from one target domain and paste onto the other target domain, and let $\mathbf{1}$ denote

the mask which is filled with one and shares the same spatial size with $\mathbf{R}$. Given the data $x^{\mathbb{T}_i}$ and $x^{\mathbb{T}_j}$ randomly sampled from $\mathbb{T}_i$ and $\mathbb{T}_j$, we have

$$x_R^{\mathbb{T}_{i,j}} = \mathbf{R} \odot x^{\mathbb{T}_i} + (\mathbf{1} - \mathbf{R}) \odot x^{\mathbb{T}_j}, \tag{7}$$

where $x_R^{\mathbb{T}_{i,j}}$ is the augmented data by mixing the regions of images randomly selected from $\mathbb{T}_i$ and $\mathbb{T}_j$, and $\odot$ is Hadmard product. The pseudo labels of $x_R^{\mathbb{T}_{i,j}}$ represented by $\hat{y}_R^{\mathbb{T}_{i,j}}$ is formulated by

$$\hat{y}_R^{\mathbb{T}_{i,j}} = \mathbf{R} \odot \hat{y}^{\mathbb{T}_i} + (\mathbf{1} - \mathbf{R}) \odot \hat{y}^{\mathbb{T}_j}, \tag{8}$$

where $\hat{y}^{\mathbb{T}_i}$ and $\hat{y}^{\mathbb{T}_j}$ are the pseudo labels of $x^{\mathbb{T}_i}$ and $x^{\mathbb{T}_j}$ generated from the EMA teacher $g_{\theta'}$. Based on the mixed images, our region-level domain interpolation builds a region-level intermediate domain $\mathbb{T}_{i,j}^R = \{x_{R,n}^{\mathbb{T}_{i,j}}\}_{n=1}^{N^{\mathbb{T}_{i,j}}}$ between $\mathbb{T}_i$ and $\mathbb{T}_j$.

With the help of image-level domain interpolation and region-level domain interpolation, our CDC congregates all the unlabeled target domains and the generated intermediate domains into a target superdomain $\mathbb{T}^{\ddagger}$ by

$$\mathbb{T}^{\ddagger} = \bigcup_{1 \leq (i,j) \leq K, i \neq j}^{K} \left( \mathbb{T}_i \cup \mathbb{T}_{i,j}^I \cup \mathbb{T}_{i,j}^R \right). \tag{9}$$

To bridge the style gap and contextual gap among target domains, we apply *mixing*-based self-supervised augmentation consistency to optimize the network $g_{\theta}$ with the losses formulated by

$$\begin{aligned} \mathcal{L}_I^{\mathbb{T}_{i,j}} &= \mathcal{F}\big(g_{\theta}(x_I^{\mathbb{T}_{i,j}})\big)\mathcal{C}\big(g_{\theta}(x_I^{\mathbb{T}_{i,j}}), \hat{y}^{\mathbb{T}_i}\big), \\ \mathcal{L}_R^{\mathbb{T}_{i,j}} &= \mathcal{F}\big(g_{\theta}(x_R^{\mathbb{T}_{i,j}})\big)\mathcal{C}\big(g_{\theta}(x_R^{\mathbb{T}_{i,j}}), \hat{y}_R^{\mathbb{T}_{i,j}}\big). \end{aligned} \tag{10}$$

### 3.4 Multi-context Masking Consistency

Despite training with *mixing*-based self-supervised augmentation consistency losses in Eq. 10, we want to enforce further alignment of the *contextual gap* among target domains. Hoyer et al. (2023) propose to improve the model's learning of image contexts with *masking*-based self-supervised augmentation consistency, which enforces consistency between the prediction of a masked image and its corresponding complete image. In this way, the model is encouraged to make semantic predictions of the masked patches based on their contextual surroundings within the image. We extend this idea and propose multi-contextual masking consistency (MCMC) to align the contextual gap among the target domains. Our MCMC applies masking consistency onto $\mathbb{T}_{i,j}^R$ to reinforce the model's learning of the *diverse contexts* of different target domains *together*.

Given a mixed image $x_R^{\mathbb{T}_{i,j}}$, we randomly mask out some patches of $x_R^{\mathbb{T}_{i,j}}$ with a patch mask $\mathbf{P}$. The patch mask is generated by using a uniform distribution with

$$\begin{aligned} \mathbf{P}_{(h,w)} &= \mathbb{1}\{\mu > \eta\}, \\ \alpha\psi \leq h &< (\alpha+1)\psi, \beta\psi \leq w < (\beta+1)\psi, \end{aligned} \tag{11}$$

where $\mu$ is sampled from a uniform distribution $\mathcal{U}(0,1)$, $\eta$ is the masking ratio, $\mathbb{1}$ is the indicator function, $\psi$ is the patch size, and $\alpha \in [0, \dots, H/\psi - 1]$ and $\beta \in [0, \dots, W/\psi - 1]$ represent the patch indices. Then we conduct element-wise multiplication of mask and image to generate the masked image $x_M^{\mathbb{T}_{i,j}}$ by

$$x_M^{\mathbb{T}_{i,j}} = \mathbf{P} \odot x_R^{\mathbb{T}_{i,j}}. \tag{12}$$

The segmentation network $g_{\theta}$ takes $x_M^{\mathbb{T}_{i,j}}$ as input and is enforced to generate the consistent predictions with the pseudo label $\hat{y}_R^{\mathbb{T}_{i,j}}$ (Eq. 8). The loss function $\mathcal{L}_M^{\mathbb{T}_{i,j}}$ for *masking*-based self-supervised augmentation consistency is denoted as

$$\mathcal{L}_M^{\mathbb{T}_{i,j}} = \mathcal{F}\big(g_{\theta}(x_M^{\mathbb{T}_{i,j}})\big)\mathcal{C}\big(g_{\theta}(x_M^{\mathbb{T}_{i,j}}), \hat{y}_R^{\mathbb{T}_{i,j}}\big) \tag{13}$$

In this way, $g_{\theta}$ is encouraged to learn the *diverse contexts* from $\mathbb{T}_i$ and $\mathbb{T}_j$ *together* to predict the correct labels for the masked out patches.

We present an overview of our MacDC framework in Fig. 3. The final loss function $\mathcal{L}^{\ddagger}$ for training MacDC is formulated by

$$\mathcal{L}^{\ddagger} = \mathcal{L}^{\mathbb{S}} + \sum_{i,j(i \neq j)}^{K} (\lambda_A \mathcal{L}_A^{\mathbb{T}_i} + \lambda_I \mathcal{L}_I^{\mathbb{T}_{i,j}} + \lambda_R \mathcal{L}_R^{\mathbb{T}_{i,j}} + \lambda_M \mathcal{L}_M^{\mathbb{T}_{i,j}}). \tag{14}$$

Table 1: Comparison of syn-to-real MTDA performance under 19 categories on adapting from GTA5 (G) to Cityscapes (C) and IDD (I).

| Method | Trg | Road | Side | Buil | Wall | Fence | Pole | TL | TS | Vege | Terr | Sky | Pers | Rider | Car | Truck | Bus | Train | Motor | Bike | mIoU | Avg |
|---|---|---|---|---|---|---|---|---|---|---|---|---|---|---|---|---|---|---|---|---|---|---|
| MTKT | C | 88.8 | 23.8 | 81.5 | 27.7 | 27.3 | 31.7 | 33.2 | 22.9 | 83.1 | 27.0 | 76.4 | 58.5 | 28.9 | 84.3 | 30.0 | 36.8 | 0.3 | 27.7 | 33.1 | 43.3 | 43.5 |
| | I | 94.1 | 24.4 | 66.1 | 31.3 | 22.0 | 25.4 | 9.3 | 26.7 | 80.0 | 31.4 | 93.5 | 48.7 | 43.8 | 71.4 | 49.4 | 28.5 | 0.0 | 48.7 | **34.3** | 43.6 | |
| CCL | C | 90.3 | 34.0 | 82.5 | 26.2 | 26.6 | 33.6 | 35.4 | 21.5 | 84.7 | 39.8 | 81.1 | 58.4 | 25.8 | 84.5 | 31.4 | 45.4 | 0.0 | 29.9 | 24.7 | 45.0 | 45.5 |
| | I | 95.0 | 30.5 | 65.6 | 29.4 | 23.4 | 29.2 | 12.0 | 37.8 | 77.3 | 31.3 | 91.9 | 52.4 | 48.3 | 74.9 | 50.1 | 36.6 | 0.0 | 56.1 | 32.4 | 46.0 | |
| ADAS | C | - | - | - | - | - | - | - | - | - | - | - | - | - | - | - | - | - | - | - | 45.8 | 46.1 |
| | I | - | - | - | - | - | - | - | - | - | - | - | - | - | - | - | - | - | - | - | 46.3 | |
| CoaST | C | 81.7 | 38.3 | 71.0 | 33.3 | 30.7 | 35.1 | 38.2 | 37.6 | 86.4 | **46.9** | 81.9 | 63.4 | 27.4 | 84.5 | 29.4 | 45.6 | **0.3** | **32.6** | 31.3 | 47.1 | 48.2 |
| | I | 85.7 | 36.1 | 65.1 | 33.2 | 23.7 | **32.8** | 19.0 | 62.9 | 82.5 | 29.5 | 91.8 | 52.1 | 55.3 | 83.4 | 62.9 | 46.1 | 0.0 | 55.5 | 18.5 | 49.3 | |
| Ours (NDC) | C | 94.8 | 62.1 | 87.2 | 36.0 | 33.1 | **38.3** | 48.7 | 48.9 | 86.9 | 39.9 | 88.9 | 61.6 | 16.4 | 89.4 | 57.6 | 63.1 | 0.0 | 15.0 | 39.6 | 53.0 | 51.4 |
| | I | 95.5 | 48.6 | 59.9 | 37.1 | 25.5 | 21.6 | 16.8 | 60.2 | 80.6 | 43.8 | **93.8** | 48.7 | 47.8 | 69.8 | 66.8 | 50.1 | 0.0 | 59.7 | 17.8 | 49.7 | |
| Ours (MacDC) | C | **95.3** | **70.3** | **89.2** | **39.4** | **35.0** | 38.1 | **53.2** | **50.3** | **87.8** | 40.2 | **90.2** | **71.2** | 46.4 | **91.4** | **61.0** | **67.3** | 0.0 | 18.7 | **45.2** | **57.4** | **56.3** |
| | I | **97.2** | **61.9** | **67.8** | **45.5** | 26.8 | 28.6 | **20.4** | **71.2** | **82.8** | **44.3** | 92.7 | **58.5** | **56.8** | **75.2** | **78.3** | **64.5** | 0.0 | **64.4** | 13.6 | **55.3** | |

Table 2: Comparison of syn-to-real MTDA performance under 7 categories on adapting from GTA5 (5) to Cityscapes (G) and IDD (I).

| Method | Trg | Flat | Const | Obje | Nature | Sky | Human | vehicle | mIoU | Avg |
|---|---|---|---|---|---|---|---|---|---|---|
| MTKT | C | 94.5 | 82.0 | 23.7 | 80.1 | 84.0 | 51.0 | 77.6 | 70.4 | 68.2 |
| | I | 91.4 | 56.6 | 13.2 | 77.3 | 91.4 | 51.4 | 79.9 | 65.9 | |
| ADAS | C | 95.1 | 82.6 | **39.8** | 84.6 | 81.2 | 63.6 | 80.7 | 75.4 | 71.2 |
| | I | 90.5 | **63.0** | 22.2 | 73.7 | 87.9 | 54.3 | 76.9 | 66.9 | |
| CoaST | C | 94.7 | 82.9 | 25.4 | 82.2 | **88.2** | 54.4 | 80.5 | 72.6 | 71.3 |
| | I | 94.2 | 61.5 | 20.0 | 82.7 | **93.4** | 55.5 | 82.6 | 70.0 | |
| Ours (NDC) | C | 94.8 | 84.8 | 31.3 | 85.5 | 81.6 | 63.2 | 84.7 | 75.1 | 72.7 |
| | I | 94.1 | 60.7 | 21.7 | 82.8 | 90.4 | 57.4 | 85.1 | 70.3 | |
| Ours (MacDC) | C | **95.4** | **85.7** | 33.8 | **88.4** | 81.5 | **68.7** | **89.4** | **77.6** | **75.5** |
| | I | **94.3** | 62.4 | **24.9** | **86.3** | 92.1 | **66.2** | **88.3** | **73.5** | |

Table 3: Summary of real-to-real MTDA performance under 19 categories with Cityscapes (C), IDD (I), and Mapillary (M).

| | Methods | mIoU C | I | M | mIoU Avg |
|---|---|---|---|---|---|
| C→I,M | CCL | - | 53.6 | 51.4 | 52.5 |
| | ADAS | - | 48.3 | 53.6 | 50.5 |
| | CoaST | - | 50.8 | 52.7 | 51.7 |
| | Ours (MacDC) | - | **56.2** | **55.4** | **55.8** |
| I→C,M | CCL | 46.8 | - | 49.8 | 48.3 |
| | ADAS | 49.1 | - | 50.8 | 50.0 |
| | CoaST | 48.9 | - | 53.4 | 51.2 |
| | Ours (MacDC) | **54.3** | - | **55.6** | **55.0** |
| M→C,I | CCL | 58.5 | 54.1 | - | 56.3 |
| | ADAS | 58.7 | 54.1 | - | 56.4 |
| | CoaST | 59.3 | 57.4 | - | 58.3 |
| | Ours (MacDC) | **63.3** | **59.4** | - | **61.4** |

## 4 EXPERIMENTS

**Datasets.** To ensure a fair comparison with existing MTDA approaches (Zhang et al., 2023; Isobe et al., 2021; Lee et al., 2022; Saporta et al., 2021), we conduct experiments involving four datasets with distinct scene structures and visual appearances. The datasets include one synthetic dataset used as the source domain and three real-world datasets used as the target domains. GTA5 (Richter et al., 2016) is a synthetic dataset of densely annotated driving images from a video game simulator. Cityscapes (Cordts et al., 2016) is an urban driving data set from Europe in clear weather. Mapillary (Neuhold et al., 2017) is a large-scale driving scene dataset collected from multiple regions around the world under various weather, season and daytime conditions. IDD (Varma et al., 2019) is an Indian driving scene dataset. Its scene structures differ from those in Mapillary and Cityscapes, resulting in unique challenges in less structured driving environments.

**Benchmarks.** We compare our proposed MacDC with existing MTDA approaches including *MTKT* (Saporta et al., 2021), *CCL* (Isobe et al., 2021), *ADAS* (Lee et al., 2022), and *CoaST* (Zhang et al., 2023). We present two MTDA schemes including *synthetic-to-real* adaptation and *real-to-real* adaptation. Following the setting of existing MTDA approaches, we conduct experiments under two category settings: 19 categories and 7 categories. The performance in each target domain is presented in the metric of mIoU (%). The *Avg* is to take the average over the multiple target domains' mIoU.

**Implementation Details.** Like other MTDA approaches, our model leverages a DeepLab-v2 network (Chen et al., 2017) with ResNet101 backbone (He et al., 2016). We optimize the model using SGD with a weight decay of $5 \times 10^{-4}$ and a momentum of $0.9$. The initial learning rate is $2.4 \times 10^{-4}$. We set the quality evaluation threshold $\xi = 0.986$. The source images and target images are first resized to $1280 \times 720$ and $1024 \times 512$ respectively, and then cropped into $512 \times 512$. We adopt color augmentation including brightness, blur, saturation, and contrast. For the multi-contextual masking consistency, we set a patch size $\psi = 64$ and a masking ratio of $\eta = 0.7$. The training process takes $40,000$ iterations with a batch size of $4$. The loss weights $\lambda_A, \lambda_I, \lambda_R, \lambda_M$ are all set with $1$. Our model is trained on a GeForce RTX 3090 GPU.

Table 4: Summary of syn-to-real MTDA performance under 19 categories with GTA5 (G), Cityscapes (C), IDD (I), and Mapillary (M).

| | Methods | mIoU C | mIoU I | mIoU M | mIoU Avg |
|---|---|---|---|---|---|
| G→C,I | CCL | 45.0 | 46.0 | - | 45.5 |
| | ADAS | 45.8 | 46.3 | - | 46.1 |
| | CoaST | 47.1 | 49.3 | - | 48.2 |
| | Ours (MacDC) | 57.4 | 55.3 | - | 56.3 |
| G→C,M | CCL | 45.1 | - | 48.8 | 47.0 |
| | ADAS | 45.8 | - | 49.2 | 47.5 |
| | CoaST | 47.9 | - | 51.8 | 49.9 |
| | Ours (MacDC) | 57.1 | - | 56.4 | 56.8 |
| G→I,M | CCL | - | 44.5 | 46.4 | 45.5 |
| | ADAS | - | 46.1 | 47.6 | 46.9 |
| | CoaST | - | 49.5 | 51.6 | 50.6 |
| | Ours (MacDC) | - | 53.6 | 55.3 | 54.4 |
| G→C,I,M | CCL | 46.7 | 47.0 | 49.9 | 47.9 |
| | ADAS | 46.9 | 47.7 | 51.1 | 48.6 |
| | CoaST | 47.2 | 48.7 | 51.4 | 49.1 |
| | Ours (MacDC) | 56.2 | 53.3 | 54.6 | 54.7 |

Table 5: Ablation study on image-level domain interpolation methods.

| Method | mIoU |
|---|---|
| NDC | 51.4 |
| NDC + CycleGAN ($\mathbb{T}_i \leftrightarrow \mathbb{T}_j$) | $52.9 \pm 0.5$ |
| NDC + ColorTransfer ($\mathbb{T}_i \leftrightarrow \mathbb{T}_j$) | $53.3 \pm 1.1$ |

Table 6: Ablation study on region-level domain interpolation methods.

| Method | mIoU |
|---|---|
| NDC | 51.4 |
| NDC + CowMix ($\mathbb{T}_i \leftrightarrow \mathbb{T}_j$) | $52.2 \pm 0.2$ |
| NDC + FMix ($\mathbb{T}_i \leftrightarrow \mathbb{T}_j$) | $51.8 \pm 0.3$ |
| NDC + CutMix ($\mathbb{T}_i \leftrightarrow \mathbb{T}_j$) | $52.5 \pm 0.3$ |

Table 7: Ablation study of each component of MacDC on adapting GTA5 to Cityscapes and IDD in 19 categories.

| Configuration | $\mathcal{L}_A^{\mathbb{T}_i}$ | $\mathcal{L}_I^{\mathbb{T}_{i,j}}$ | $\mathcal{L}_R^{\mathbb{T}_{i,j}}$ | $\mathcal{L}_M^{\mathbb{T}_{i,j}}$ | Avg | Gain |
|---|---|---|---|---|---|---|
| Naive Domain Congregation (NDC) | ✓ | | | | 51.4 | - |
| NDC + Image-level Domain Interp. | ✓ | ✓ | | | 53.3 | +1.9 |
| NDC + Region-level Domain Interp. | ✓ | | ✓ | | 52.5 | +1.2 |
| NDC + Multi-context. Mask. Consis. | ✓ | | ✓ | ✓ | 54.5 | +3.2 |
| Collabora. Domain Congregation | ✓ | ✓ | ✓ | | 53.7 | +2.4 |
| *Full Framework* | ✓ | ✓ | ✓ | ✓ | 56.3 | +4.9 |

Table 8: Training time and memory consumption analysis.

| Method | Train Time | GPU |
|---|---|---|
| MTKT | 39.7 hr | 13.2 GB |
| CoaST | 105.3 hr | 17.5 GB |
| Ours (NDC) | 32.1 hr | 16.3 GB |
| Ours (MacDC) | 32.5 hr | 16.8 GB |

## 4.1 Comparison with State-of-the-Art Methods

**Synthetic-to-Real Adaptation.** We conduct experiments in the setting of synthetic-to-real adaptation setting following the existing MTDA methods. The source domain is GTA5, and the multiple target domains are Cityscapes, IDD, and Mapillary. The results are presented in a comprehensive comparison with existing MTDA methods on adapting GTA5 to Cityscapes and IDD under 19 categories in Table 1 and 7 categories in Table 2. Note that *Avg* is to take the average over all target domains' mIoU. Despite its simplicity, our NDC outperforms state-of-the-art MTDA methods. This is owing to the usage of ClassMix as data augmentation in NDC in comparison to adversarial training in existing MTDA methods. In addition, our MacDC is present to mitigate the style and contextual gaps among target domains and further improve the performance of NDC. We further summarize the synthetic-to-real MTDA performance with various combinations of target domains in Table 4. Our MacDC demonstrates persistent superior performance against existing methods. The qualitative results of our MacDC are shown in Fig. 4.

**Real-to-Real Adaptation.** We further study the scalability of our approach through the setting of real-to-real adaptation scheme. Given the three datasets Cityscapes, IDD, and Mapillary, we take one as the source domain and the other two as the target domains. We conduct experiments on real-to-real adaptation in 19 categories and summarize the results in Table 3. Note that *Avg* is to take the average over all target domains' mIoU. Overall the cases in real-to-real adaptation, our MacDC advances the performance in comparison with existing methods CCL, ADAS, and CoaST. It shows that our MacDC as a simple and effective MTDA approach, can be used in not only the synthetic-to-real but also the real-to-real adaptation setting.

## 4.2 Ablation Study

**NDC.** We compare the performance of naive domain congregation (NDC) with state-of-the-art MTDA methods. The ablation study is conducted on GTA5 to Cityscapes and IDD. NDC outperforms current best MTDA CoaST by $3.2\%$ mIoU in 19 categories (Table 1) and $1.4\%$ mIoU in 7 categories (Table 2). For the MTDA semantic segmentation task, this ablation study indicates that

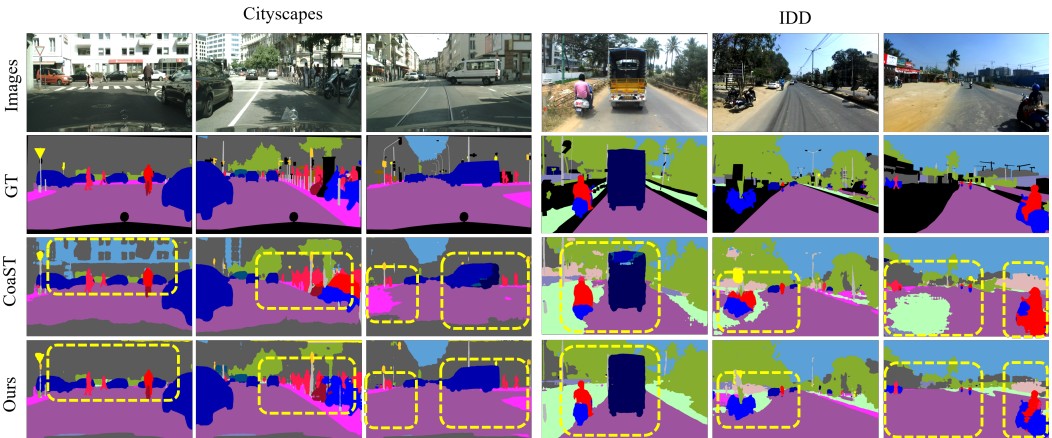

Figure 4: Qualitative results of CoaST and our MacDC for adapting GTA5 to Cityscapes and IDD under 19 categories.

self-supervised augmentation consistency, such as ClassMix, is more effective than the adversarial learning adopted in CoaST.

**CDC and MCMC.** The ablation study on the effectiveness of collaborative domain congregation (CDC) and multi-context masking consistency (MCMC) is illustrated in Table 7. While using only image-level domain interpolation, the model achieves $1.9\%$ higher in mIoU compared with NDC. Similarly, using only region-level domain interpolation brings $1.2\%$ boost. The usage of CDC which combines image-level and region-level domain interpolation gives an improvement of $2.4\%$. In contrast, utilizing MCMC delivers $3.2\%$ performance gain as MCMC can effectively promote the model's understanding of diverse contextual information among the target domains.

**Domain Interpolation Methods.** We also conduct ablation study on different domain interpolation methods for collaborative domain congregation. We first compare the performance of utilizing CycleGAN and ColorTranfer in image-level domain interpolation shown in Table 5. Furthermore, we compare CowMix, FMix, and CutMix in region-level domain interpolation presented in Table 6. We adopt ColorTransfer for image-level interpolation and CutMix for region-level interpolation as they perform slightly better than their counterparts.

**Training Time and Memory Consumption.** The ablation study of training time and memory requirement is illustrated in Table 8. Though MTKT requires moderate training time and less GPU memory, it delivers limited performance in MTDA for segmentation. While CoaST presents higher scores compared with MTKT, it requires much longer time for training (multiple stages). In comparison, our MacDC requires moderate GPU memory and much less training time, at the same time delivering superior performance against CoaST and MTKT.

## 5 CONCLUSION

This paper addresses the challenges in multi-target domain adaptation for semantic segmentation task. Existing methods shows limited performance as they only consider the style difference while ignoring the contextual variations among these target domains. In contrast, we propose a novel approach named masking-augmented collaborative domain congregation to address the style gap and the contextual gap altogether. We handle the style and contextual gaps among target domains by data mixing, which generates image-level and region-level intermediate domains among target domains. We further enforces the model's understanding of diverse contexts together with a masking consistency. Our proposed MacDC effectively mitigates style and contextual gaps among multiple target domains by directly learning a single model for multi-target domain adaptation without requiring multiple network training and subsequent distillation. The experimental results on MTDA for segmentation benchmarks highlight the effectiveness of our approach.

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
