# OpenReview forum: "MacDC: Masking-augmented Collaborative Domain Congregation for Multi-target Domain Adaptation in Semantic Segmentation"
_ICLR.cc/2024/Conference — ICLR 2024 Conference Withdrawn Submission_

### Official Review · Reviewer_zRcv · 2023-10-30

**Soundness:** 2 fair
**Presentation:** 3 good
**Contribution:** 2 fair
**Rating:** 3
**Confidence:** 5

**Summary:**

The paper primarily focuses on the domain-adaptive semantic segmentation task for multiple target domains. The authors introduce a so-called "Masking-augmented Collaborative Domain Congregation" (MacDC) to address both style and context disparities concurrently. MacDC comprises the "Collaborative Domain Congregation" (CDC) and the "Multi-Context Masking Consistency" (MCMC) techniques. The CDC generates two levels of intermediary domains through style transfer techniques and data-mixing, while MCMC further models masked regions in the data-mixed intermediary domain to allow the model to grasp diverse contextual scenarios.

**Strengths:**

1. The exposition on related techniques is quite comprehensive.

2. The ablation studies are thorough, and the experimental settings are clearly presented.

3. The writing in this paper is commendable and easy to follow.

**Weaknesses:**

This paper appears to be a compilation of existing techniques in the UDA semantic segmentation field. Techniques like style transfer, data-mixing, and masked image modeling are already widely adopted in UDA semantic segmentation. The authors haven't introduced any unique components specifically for the MTDA problem. I suggest that the authors apply well-established UDA semantic segmentation techniques directly in the MTDA setting, such as HRDA [A] and MIC [B], and include their results in the table for comparison. I believe these UDA semantic segmentation techniques should be considered for a fair comparison.


[A] HRDA: Context-aware high-resolution domain adaptive semantic segmentation. ECCV, 2022
[B] MIC: Masked image consistency for context-enhanced domain adaptation. CVPR, 2023

**Questions:**

See weaknesses above.

---

### Official Review · Reviewer_TU98 · 2023-10-30

**Soundness:** 3 good
**Presentation:** 3 good
**Contribution:** 2 fair
**Rating:** 3
**Confidence:** 4

**Summary:**

This paper presents a strong baseline and a new method for multi-target domain adaptive semantic segmentation task. By taking the multi-target domain as a single domain and applying classmix and mean teacher framework like semi-supervised segmentation, this paper firstly presents a strong baseline with the performance on par with the state-of-the-art method. Then, the paper proposes the idea of Collaborative domain Congregation to mitigate the style gap and contextual gap between multiple target domains. Specifically, the Collaborative domain Congregation consists of image-level domain interpolation for creating intermediate target domain with style transfer and Region-level Domain Interpolation for merging contents from different domains with data mixing methods. Plus, to promote the contextual gap alignment, this paper applies the multi-context masking consistency to multiple domains for enforcing the consistency of predictions on masked images and the pseudo labels. Extensive experiments on various benchmarks present the proposed method outperform the state-of-the-art methods in accuracy and training time.

**Strengths:**

1. The paper is well written and the contributions of this paper are presented clearly.
2. The proposed method outperforms the existing methods in accuracy by a large margin. The proposed NDA is also a strong baseline, which could benefit further works.

**Weaknesses:**

1. The proposed Collaborative domain Congregation seems to be a concept of combining style transfer module and data mixing methods. Although it may be new in multi-target domain adaptive semantic segmentation, it is well studied in the related fields like domain adaptive semantic segmentation[1][2]. As no new module is proposed for Collaborative domain Congregation, the contribution seems to be limited.
2. The proposed multi-context masking consistency(MCMC) seems to simply apply the MIC loss proposed in [3] for single domain to multi-domains. As MCMC apply the same MIC loss for multiple domains, it is like to treat all target domains(original target domains+intermediate target domains) as a single domain to perform masking consistency loss. From my view, limited insights for multi-domain adaptation is reflected in this design.
3. The analysis of the proposed method is limited. Although the ablation study presents the proposed modules could bring performance improvement, some ablation studies to prove whether the improvement is from better context utilization is expected.
4. The proposed method seems to be compatible to general multi-target domain adaptation(MTDA) for various vision tasks, while it is only verified on semantic segmentation task. I believe applying the proposed method on more MTDA tasks could bring more contributions.
5. The proposed method is only verified on ResNet101 based DeeplabV2, which is a quite old architecture for semantic segmentation.

[1] Wilhelm Tranheden, Viktor Olsson, Juliano Pinto, Lennart Svensson. DACS: Domain Adaptation via Cross-domain Mixed Sampling. WACV 2021.

[2] Lukas Hoyer, Dengxin Dai, Luc Van Gool. DAFormer: Improving Network Architectures and Training Strategies for Domain-Adaptive Semantic Segmentation. CVPR 2022.

[3] Lukas Hoyer, Dengxin Dai, Haoran Wang, Luc Van Gool. MIC: Masked Image Consistency for Context-Enhanced Domain Adaptation. CVPR 2023

**Questions:**

1. The image-level domain interpolation is just to use transfer learning to transfer the image from one target domain to another target domain. The image style after style transfer is still the styles existing in the given multiple target domains. Is it possible to use domain interpolation methods like DLOW[1] to sample a new style between two target domains? Could it further improve the performance?
2. Do you perform ablation study on patch size and masking ratio for multi-context masking consistency? Is the current value optimal for MTDA?
3.  Does the training time contain the image-level domain interpolation process?
4. Do you consider to apply your method to normal-to-adverse adaptation setting? As there are many dataset for urban scene under adverse conditions like Night Driving[2], Dark Zurich[3], ACDC[4], it is possible to benchmark your method on this setting to be aligned with domain adaptive semantic segmentation?

[1] Rui Gong, Wen Li, Yuhua Chen, Luc Van Gool. DLOW: Domain Flow for Adaptation and Generalization. CVPR 2019

[2] Dengxin Dai, Luc Van Gool. Dark Model Adaptation: Semantic Image Segmentation from Daytime to Nighttime. ICITS 2018

[3] Christos Sakaridis, Dengxin Dai, Luc Van Gool. Map-Guided Curriculum Domain Adaptation and Uncertainty-Aware Evaluation for Semantic Nighttime Image Segmentation. T-PAMI

[4] Christos Sakaridis, Dengxin Dai, Luc Van Gool. ACDC: The Adverse Conditions Dataset with Correspondences for Semantic Driving Scene Understanding. ICCV 2021

---

### Official Review · Reviewer_9HYr · 2023-11-02

**Soundness:** 2 fair
**Presentation:** 3 good
**Contribution:** 2 fair
**Rating:** 5
**Confidence:** 4

**Summary:**

This paper proposes to handle the multi-target domain segmentation learning via combining the style and the contextual cues. Specifically, a color-wise image style alignment is adopted for the former one, while a region-level interpolation with masking is adopted for the later one. Experimental evaluation shows the effectiveness of the proposed method.

**Strengths:**

+ The claim of reducing the domain-wise gap via style and context interpolation is important and reasonable. The following masking consistency unit is another novel point to ensure the semantic prediction to be generalized across different target domains.
+ The overall experimental evaluation is solid, while the comparison with SOTA methods is not sufficient. More SOTA methods should be added

**Weaknesses:**

- The overall pipeline is a combination of existing methods, i.e., the masking consistency is borrowed from the MAE [1], the class mixing is from [Chen et al. 2022], and the region-level interpolation from CowMix, FMix, and CutMix. Besides the adopted units, rare new contribution is recognized, making the overall novelty to be somewhat limited.
[1] Masked Autoencoders Are Scalable Vision Learners. 2021.

- The experimental comparisons with current SOTA methods are not sufficient. There are only four methods have been surveyed and compared.

- The lambda parameters in Eq14 are all set to 1 for any datasets? If so, there is no need to add this weighting terms in Eq14.

**Questions:**

See weakness.

---

### Meta-Review · Area_Chair_HMm4 · 2023-12-05

**Metareview:**

Three experts reviewed the paper, and none was supportive. There was no rebuttal.

**Justification For Why Not Higher Score:**

No reviewer was supportive.

**Justification For Why Not Lower Score:**

No lower score available.

---

### Decision · Program_Chairs · 2024-01-16

Reject